# Assessment of the Impacts of Land Use/Cover Change and Rainfall Change on Surface Runoff in China

**Fazhi Li** [1], **Jingqiu Chen** [2], **Yaoze Liu** [3], **Peng Xu** [4], **Hua Sun** [1,*], **Bernard A. Engel** [2,*] **and Shizhong Wang** [5]

1   College of Public Administration, Nanjing Agricultural University, Nanjing 210095, China
2   Department of Agricultural and Biological Engineering, Purdue University, IN 47907, USA
3   Department of Environmental and Sustainable Engineering, University at Albany, NY 12222, USA
4   Meteorological service center of Henan province, Zhengzhou 450003, China
5   Guangdong Provincial Key Laboratory of Environmental Pollution Control and Remediation Technology, Sun Yat-sen University, Guangzhou 510275, China
*   Correspondence: sh@njau.edu.cn (H.S.); engelb@purdue.edu (B.A.E.);
    Tel.: +86-137-7067-6369 (H.S.); +1-765-494-8362 (B.A.E.)

**Abstract:** Assessment of the impacts of land use/cover change (LUCC) and rainfall change on surface runoff depth can help provide an understanding of the temporal trend of variation of surface runoff and assist in urban construction planning. This study evaluated the impacts of LUCC and rainfall change on surface runoff depth by adopting the well-known Soil Conservation Service-Curve Number (SCS-CN) method and the widely used Long-Term Hydrologic Impact Assessment (L-THIA) model. National hydrologic soil group map of China was generated based on a conversion from soil texture classification system. The CN values were adjusted based on the land use/cover types and soil properties in China. The L-THIA model was configured by using the adjusted CN values and then applied nationally in China. Results show that nationwide rainfall changes and LUCC from 2005 to 2010 had little impact on the distribution of surface runoff, and the high values of runoff depth were mainly located in the middle and lower reaches of the Yangtze River. Nationally, the average annual runoff depths in 2005, 2010 and 2015 were 78 mm, 83 mm and 90 mm, respectively. For the 2015 land use data, rainfall change caused the variation of surface runoff depth ranging from −203 mm to 476 mm in different regions. LUCC from 2005 to 2015 did not cause obvious change of surface runoff depth, but expansion of developed land led to runoff depth increases ranging from 0 mm to 570 mm and 0 mm to 742 mm from 2005 to 2010 and 2010 to 2015, respectively. Potential solutions to urban land use change and surface runoff control were also analyzed.

**Keywords:** land use/cover change; rainfall change; developed land; runoff depth; L-THIA model

## 1. Introduction

Urbanization in China has attracted wide international attention in recent years [1]. Urbanization typically refers to the processes in which large amounts of agricultural or other non-urban land are transformed into developed land (including the low density developed land in rural areas as well as medium and high density developed land in urban areas) for the development of society, and the land use/cover types are changed significantly [2,3]. An increase in developed land usually means growth of impervious surface area, which is regarded as the direct cause of urban surface runoff [4–6]. Urban flooding occurred on average in 185 cities per year from 2010 to 2016 in China; especially, in 2016, a total of 192 cities above the county level suffered floods, which caused direct economic losses amounting to 548.49 billion dollars [7]. In addition, due to the increase of impermeable

surfaces, surface pollutants enter water bodies with surface runoff, which increases the risk of non-point source pollution and leads to water quality degradation [8–11]. Apart from the increased impervious surface area, another main cause of urban flooding is rainfall change. Research shows that precipitation in the eastern part of China presents a large increase, while a large decrease occurred in the central region and a small increase occurred in the west from 1961 to 2010 [12].

The identification of spatial distribution and variation characteristics of surface runoff is of great practical significance for surface runoff management. Assessment of the impacts of land use/cover change (LUCC) and rainfall change on surface runoff has received increased attention in recent years [13–20]. Numerous studies assessing the effect of land use change, climate change, and urbanization expansion on surface runoff have been carried out worldwide with computer models such as Soil and Water Assessment Tool (SWAT), MIKE System Hydrological European (MIKE-SHE), Hydrological Land Use Change (HYLUC), Long-Term Hydrology Impact Assessment (L-THIA), and Storm Water Management Model (SWMM)) [17,21–31]. Among these, the L-THIA model [32] is easy to use and is a Soil Conservation Service-Curve Number (SCS-CN)-based [33] model; it relies on readily available data and performs well in the simulation of hydrology both on macroscopic and microscopic scales [34–39]. For example, Bhaduri et al. [35] assessed the impact of land use change and climate change on surface runoff volume with the L-THIA model in a small watershed located in Indiana, U.S., and concluded that an 18% increase in urban or impervious areas could lead to an estimated 80% increase in annual average runoff volume between 1973 and 1991. Chen et al. [30] quantified the urbanization impacts on surface runoff of the contiguous United States from 2001 to 2011 by developing a tabular version of L-THIA model, which was designed to expedite calculations over diverse geographical areas, and concluded that urban expansion and intensification were driving forces for surface runoff change and urbanization from 2001 to 2011 contributed 10% increase in average annual runoff volume nationally. Chen et al. [38] simulated the average annual surface runoff depths of the Great Lakes Region, USA, from 2001 to 2011, and identified the areas that had high increased annual runoff depths. Li et al. [39] used the L-THIA-LID model to evaluate the implementation impacts of green infrastructure on surface runoff in a small watershed in Michigan, U.S., and concluded that surface runoff volume could be reduced by 68% at most through implementing green infrastructure.

CN-based models are not widely used in China currently because the land use type and soil texture type classification systems in China differ from those in the U.S. Some Chinese researchers have made attempts to use the CN method to assess surface runoff through calibrating the CN values [14–16,40,41], but research areas were almost always small watersheds or single cities, and research methods are often based on complicated models. No research to date was found to use the L-THIA model to assess the impacts of LUCC and rainfall change on surface runoff depth at a national scale in China.

Assessment of the impacts of LUCC and rainfall change on surface runoff depth at a macroscopic level can help understanding the spatial-temporal distribution and variation of surface runoff depth in China, and help land use planning and urban flood management. In this study, hydrological soil group map of China was firstly built based on the reclassified soil texture types; the CN values of each land use type in China were defined and adjusted to serve as input data of L-THIA model. The impacts of LUCC and rainfall change on surface runoff depth from 2005 to 2015 were then evaluated. Additionally, the impacts of isolated LUCC and isolated rainfall change on surface runoff depth were also evaluated.

## 2. Materials and Methods

### 2.1. Input Data

Land use/cover datasets with a spatial resolution of 300 m in 2005, 2010, and 2015 were based on the research output of the European Space Agency (ESA) Climate Change Initiative (CCI) – Land Cover project. Soil mechanical properties data sets with an spatial resolution of 1000 meters (including the data set classified based on the percentages of sand, silt, and clay in the soil) and the geo-political

database provided by the Data Center for Resources and Environmental Sciences, Chinese Academy of Sciences (REDC). Daily rainfall data from 2003 to 2017 were provided by the National Centers for Environmental Information, of which 635 equally distributed rainfall monitoring sites nationwide with persistent rainfall records out of 2427 rainfall monitoring sites were chosen as the input data. Daily discharge data of the selected watersheds in 2016 were provided by 25 hydrologic stations. Figure 1 shows the distribution of rainfall monitoring stations and average annual rainfall depths from 2003 to 2017, and percentage developed of each city in 2015. Rainfall data of the 635 stations represents the rainfall distribution across the country, with more rainfall concentrated in the south, while the west and north generally receive limited rainfall. Distribution of the developed land in China is considerably unbalanced and is mainly located in the eastern coastal areas and northeast China, while developed land area in the central and western regions is relatively small.

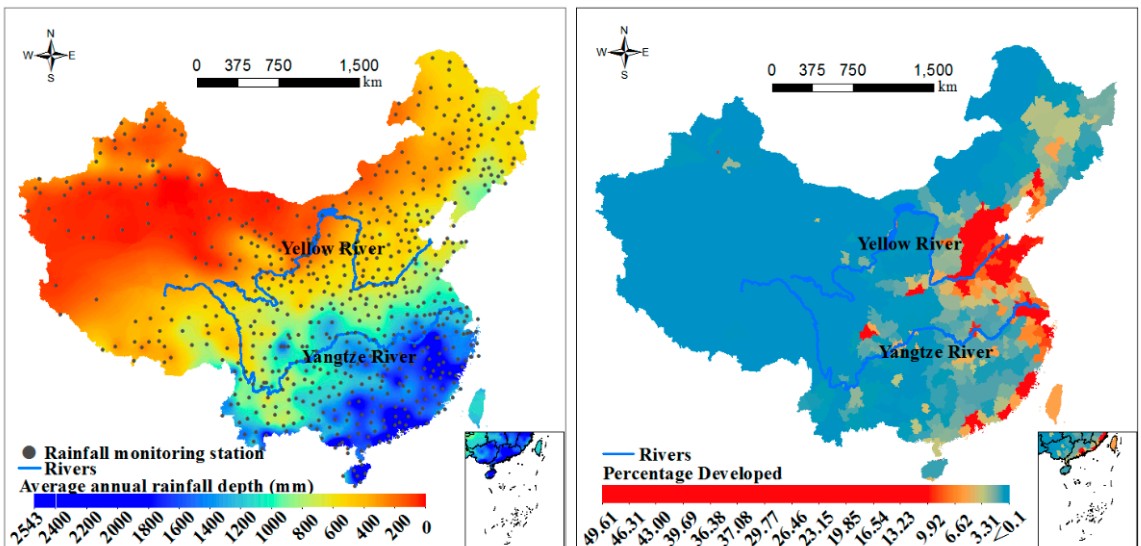

**Figure 1.** Average annual rainfall depths from 2003 to 2017 (**left**) and percentage developed in each city in 2015 (**right**).

*2.2. Research Methods*

2.2.1. Generation of Hydrologic Soil Group Map

Hydrologic soil group (HSG) classification in the USA is based on the infiltration rate, which is controlled by the soil profile [33]. Technical Release 55 (TR-55) of urban hydrology for small watersheds presents the hydrologic soil group type according to the surface soil texture [33]. Of which, the sand, loamy sand, and sandy loam belongs to type A of HSG, the silt loam and loam belongs to type B of HSG, the sandy clay loam belongs to type C of HSG, and the clay loam, silty clay loam, sandy clay, silty clay, and clay belongs to type D of HSG.

The soil texture classification system was developed by the United States Department of Agriculture (USDA) with the texture triangle diagram (Figure 2). Three legs of the equilateral triangle represent the percentages of the weights of clay (with equivalent grain size less than 0.002 mm), silt (with equivalent grain size range from 0.002 mm to 0.05 mm), and sand (with equivalent grain size range from 0.05 mm to 2 mm) [42]. One difference between the Chinese and USA soil mechanical systems is the equivalent grain size of sand, which is from 2 mm to 0.05 mm in the USA, while it is 1 mm to 0.05 mm in China (Table 1). This could decrease the accuracy of the definition of soil texture types when the textural classes were reclassified according to USA soil mechanical analysis systems using the Chinese soil mechanical data sets. However, the classification of soil texture depends on the dominant type of soil particles (Figure 2). Considering the soil particle size range of silt and clay are the same both in the Chinese and USA soil mechanical analysis systems, the different particle size range of sand was

considered acceptable. Therefore, the soil mechanical data sets were first converted to vector data sets, and then overlay analysis tools and data selection tools in ArcGIS software were used to obtain the soil texture map according to the triangular diagram. Then, the hydrologic soil group map was generated according to Technical Release 55 (Figure 3).

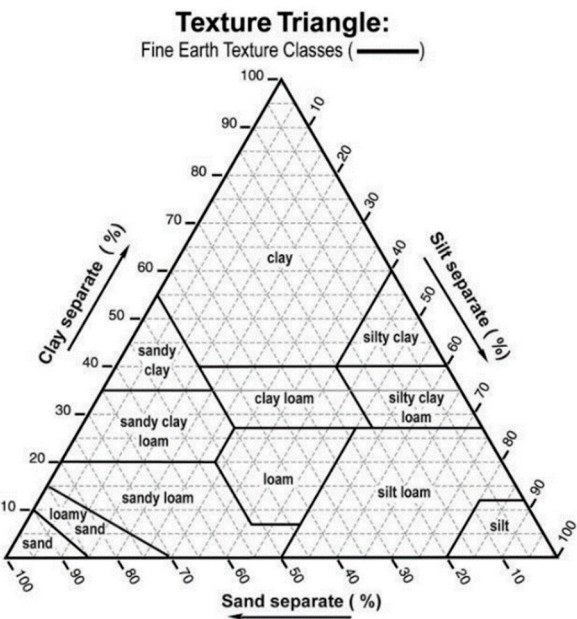

**Figure 2.** Soil texture triangular diagram of United States Department of Agriculture [42].

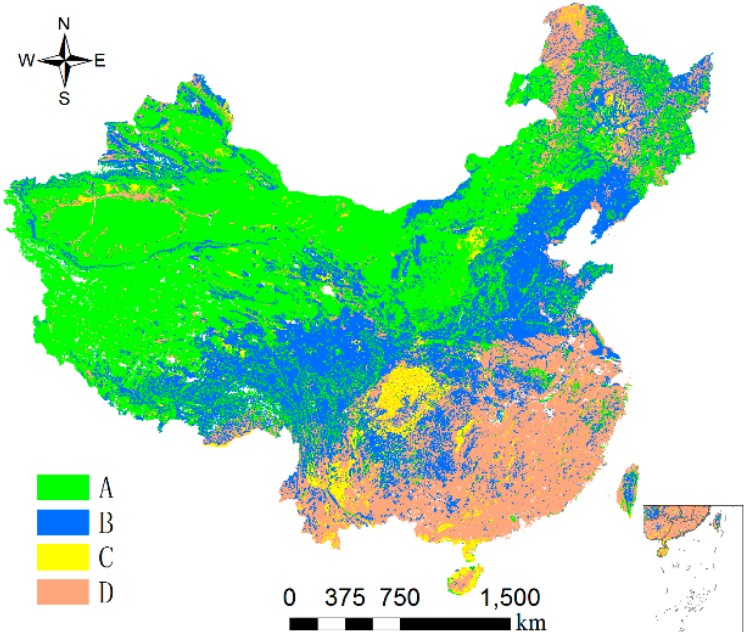

**Figure 3.** Hydrologic soil group map of China.

**Table 1.** Typical soil mechanical properties in the USA and China [43].

| Conventional Names | Equivalent Grain Size (Diameter) (mm) | |
|:---:|:---:|:---:|
| | Soil Mechanical in the USA | Soil Mechanical in China |
| Gravel | 3–2 | 3–1 |
| Sand | 2–0.05 | 1–0.05 |
| Silt | 0.05–0.002 | 0.05–0.002 |
| Clay | Less than 0.002 | Less than 0.002 |

2.2.2. ArcL-THIA10.1 Tool

The ArcL-THIA10.1 tool (Purdue University, West Lafayette, US) is a toolbox that can be used in ArcGIS version 10.1 (Environmental Systems Research Institute, Redlands, US). It was developed for the L-THIA model using Python programming language [32]. The basic principle of this model is to use the SCS-CN method to calculate runoff depth based on land use data, hydrologic soil group and daily precipitation data [44–46]. The SCS model assumes the ratio of actual retention after runoff begins (F) to actual runoff (Q) in a catchment equals the ratio of potential maximum retention after runoff begins (S) to potential maximum runoff ($Q_m$), which means:

$$\frac{F}{Q} = \frac{S}{Q_m} \tag{1}$$

$$Q_m = P - I_a, \ F = P - I_a - Q \tag{2}$$

$$Q = \frac{(P - I_a)^2}{(P - I_a) + S} \tag{3}$$

where F is actual retention after runoff begins (mm), Q is actual runoff depth (mm), S is potential maximum retention after runoff begins which depends on the CN value, $Q_m$ is potential maximum runoff depth (mm), P is rainfall depth (mm), and $I_a$ is initial abstraction [33]. $I_a$ equals to 0.2S in the L-THIA model (Following the formulation of the SCS model), which means:

$$Q = \frac{(P - 0.2S)^2}{0.8S + P} \ \ (Q = 0 \text{ for } P \le 0.2S) \tag{4}$$

$$S = \frac{25400}{CN} - 254 \tag{5}$$

The ArcL-THIA10.1 tool contains four main components. The first component is used for generating the "CN map" through overlaying land use data, hydrologic soil group data, and the CN-value table (Table 2). Each cell is created with a CN value according to the combination of land use type and hydrologic soil group. Hydrology runoff unit codes are also generated for each cell that receives the same attributes. The second component is used for generating the "rainfall allocation map" according to the coordinates of rainfall monitoring sites and the research area using the Thiessen polygon method, which determines the rainfall depth of each cell. The third component is used for generating the "CN map for Multi-Rainfall data", based on the "CN map" and the "rainfall allocation map", each hydrology runoff unit is defined with a rainfall monitoring site code for calculating the runoff depth. The fourth component is used for calculating the surface runoff volume in each hydrology unit with the CN value, daily rainfall depth and area of each cell.

Pixels with same attributes were assigned as one hydrology unit in this tool, thus it is incapable of calculating the runoff volumes of each assessment unit (an administrative division level in China, which were called cities in this study for simplicity). To solve this problem, a numbering system was used. Assessment units were first numbered from 1 to 345 (there were 345 cities in total), and each cell was multiplied by 10,000,000 to obtain an eight-digit to ten-digit value, with format of "A0000000", where A is the code of each city. Then, the land use data were rebuilt by overlaying the city data sets

and the land use data to obtain new land use type values with a format of "A00000DD", where DD is the initial land use type value. The CN value of each "new land use/cover type" was also defined according to initial land use types. The format of each hydrology runoff unit code in the "CN map" would then become "ABBBCCDD" (BBB is the code of rainfall monitoring sites and CC is the HSG code), which makes it possible to calculate the runoff volume in each city and each province through extracting the city codes. Figure 4 shows the detailed data processing steps and the principle of the L-THIA model [47].

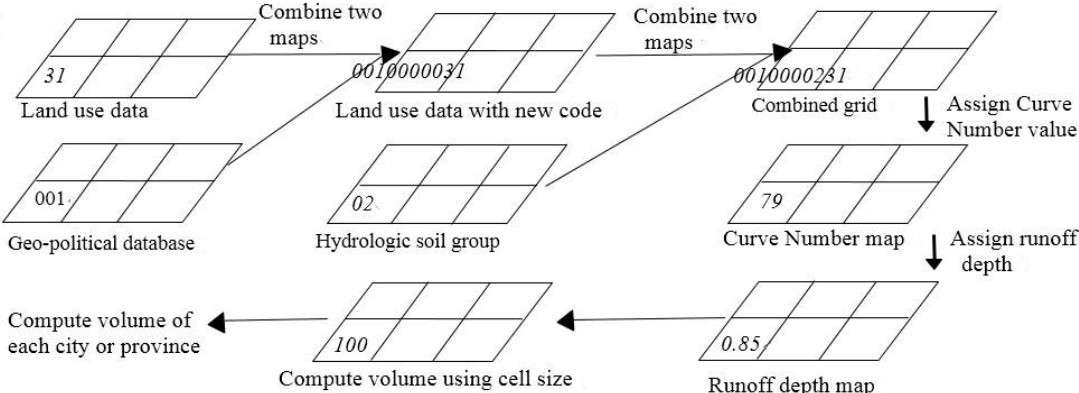

**Figure 4.** Framework of Long-Term Hydrology Impact Assessment (L-THIA) model.

### 2.2.3. CN Value Definition

The combination of land use/cover type and HSG was defined with a CN value in L-THIA model, but the land use/cover types are based on the US land use/cover type classification system [48]. However, the typology of land cover dataset in this study was defined using the Land Cover Classification System (LUCCS) developed by the United Nations (UN) Food and Agriculture Organization (FAO) [49]. Influenced by the spatial resolution of the dataset, some land use types were mixed types, percentage of each land use type in each pixel was described in detail in the product user guide [50]. CN values of pure land use types (single land use type in each pixel) were defined as the same as the US land use/cover types. CN values of mixed land use types were defined by the weighted values (Table 2).

Perennial ice/snow is mainly located in the northwest areas with high altitude, low temperature, relatively small rainfall depth, and small area (0.7% of the total area). Surface runoff generation process of perennial ice/snow is different from other land use types and the relative research regarding this topic is rare. Therefore, surface runoff generated from perennial ice/snow was not included in this study and CN value of perennial ice/snow was then defined as zero. In addition, this study is to evaluate surface runoff from the ground, and the observed surface runoff volumes used for model calibration and validation were also estimated surface runoff from the ground by applying the Baseflow Filter Program (BFLOW) [51]. Therefore, CN value of water surface was also defined as zero in this study. Detailed CN-value calculation rules are listed in Table 2. The initial values of each land use type (two to three digits values) were reclassified using 01 to 37 to meet the model requirements (two-digit values).

**Table 2.** Initial definition method of CN values based on the initial land use dataset.

| Reclassified Values of Land Use Types | Initial Values of Initial Land Use Dataset | Calculate Methods of CN Values | Reclassified Values of Land Use Types | Initial Values of Initial Land Use Dataset | Calculate Methods of CN Values |
|---|---|---|---|---|---|
| 1 | 10 | $CN_{cul.}$ | 20 | 120 | $CN_{For.}$ |
| 2 | 11 | $(CN_{cul.} + CN_{Gra.})/2$ | 21 | 121 | $CN_{For.}$ |
| 3 | 12 | $(CN_{cul.} + CN_{For.})/2$ | 22 | 122 | $CN_{For.}$ |
| 4 | 20 | $CN_{cul.}$ | 23 | 130 | $CN_{Gra.}$ |
| 5 | 30 | $0.75 \times CN_{Cul.} + 0.25 \times CN_{For.}$ | 24 | 140 | $CN_{Lic.}$ |
| 6 | 40 | $0.25 \times CN_{Cul.} + 0.75 \times CN_{For.}$ | 25 | 150 | $0.1 \times CN_{For.} + 0.9 \times CN_{Bar.}$ |
| 7 | 50 | $0.9 \times CN_{For.} + 0.1 \times CN_{Gra.}$ | 26 | 151 | $0.1 \times CN_{For.} + 0.9 \times CN_{Bar.}$ |
| 8 | 60 | $0.9 \times CN_{For.} + 0.1 \times CN_{Gra.}$ | 27 | 152 | $0.1 \times CN_{For.} + 0.9 \times CN_{Bar.}$ |
| 9 | 61 | $0.7 \times CN_{Cul.} + 0.3 \times CN_{Gra.}$ | 28 | 153 | $0.1 \times CN_{Gra.} + 0.9 \times CN_{Bar.}$ |
| 10 | 62 | $0.3 \times CN_{For.} + 0.7 \times CN_{Gra.}$ | 29 | 160 | $CN_{Wet.}$ |
| 11 | 70 | $0.9 \times CN_{For.} + 0.1 \times CN_{Gra.}$ | 30 | 170 | $CN_{Wet.}$ |
| 12 | 71 | $0.7 \times CN_{Cul.} + 0.3 \times CN_{Gra.}$ | 31 | 180 | $CN_{Wet.}$ |
| 13 | 72 | $0.3 \times CN_{For.} + 0.7 \times CN_{Gra.}$ | 32 | 190 | $CN_{Bar.}$ |
| 14 | 80 | $0.9 \times CN_{For.} + 0.1 \times CN_{Gra.}$ | 33 | 200 | $CN_{Bar.}$ |
| 15 | 81 | $0.7 \times CN_{Cul.} + 0.3 \times CN_{Gra.}$ | 34 | 201 | $CN_{Bar.}$ |
| 16 | 82 | $0.3 \times CN_{For.} + 0.7 \times CN_{Gra.}$ | 35 | 202 | $CN_{Unc.}$ |
| 17 | 90 | $CN_{For.}$ | 36 | 210 | $CN_{Wat.}$ |
| 18 | 100 | $0.75 \times CN_{Cul.} + 0.25 \times CN_{For.}$ | 37 | 220 | $CN_{Per.}$ |
| 19 | 110 | $0.25 \times CN_{Cul.} + 0.75 \times CN_{For.}$ | | | |

Note: 1 $CN_{Cul.}$ is the CN value of cultivated land, $CN_{Gra.}$ is the CN value of grassland, $CN_{For.}$ is the CN value of forest, $CN_{Bar.}$ is the CN value of barren land, $CN_{Wet.}$ is the CN value of wetland, $CN_{Unc.}$ is the CN value of unconsolidated shore and $CN_{Per.}$ is the CN value of perennial ice/snow.

2.2.4. Model Calibration and Validation

Small changes in CN values of each land cover type may cause large change in simulation results [45,52]. Calculation of CN values based on simple relationships between two kinds of land use/cover remotely sensed data classification systems may cause errors. In addition, in a country featuring a huge variation of characteristics, CN values of same kinds of land use/cover types may differ greatly in different areas. Therefore, 25 randomly distributed watersheds throughout the country were selected for model calibration (18 watersheds) and validation (7 watersheds) (Figure 5).

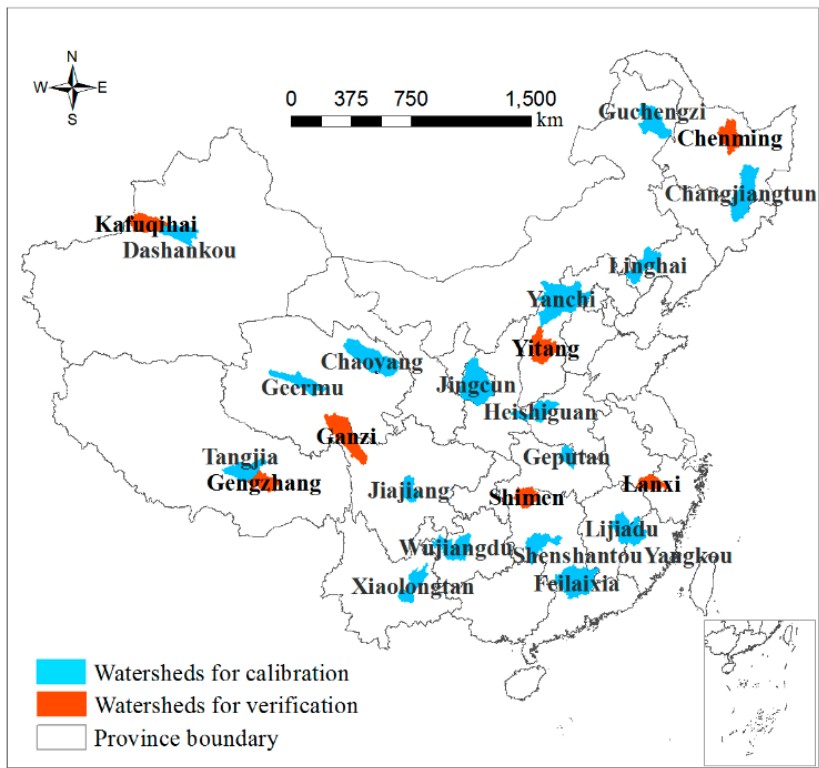

**Figure 5.** Location of the selected watersheds for model calibration and validation.

(1) Model Calibration

First, observed surface runoff volumes of the selected watersheds in 2016 were estimated from daily discharge data by applying the Baseflow Filter Program (BFLOW) [51]. Second, surface runoff volumes of 18 watersheds in 2016 were simulated with the initial L-THIA model. CN values of each watershed were adjusted by 1% each time until the simulated values matched the observed values well, and an adjustment parameter was identified for each watershed (Table 3). Considering that CN values of impervious surface equal to 98 [53–55] and the CN values of natural land surface were not bigger than that of impervious surfaces, any CN values greater than 98 after the adjustment were defined to be 98. The $R^2$ and NSE (Nash–Sutcliffe efficiency coefficient [56]) were calculated as 0.95 and 0.94, respectively, after model calibration.

**Table 3.** Surface runoff volumes of the study watersheds.

| Calibration or Validation | Hydrologic Station | Longitude (°) | Latitude (°) | Watershed Area (km$^2$) | Observed Runoff Volume (1 × 10$^8$ m$^3$) | Simulated Runoff Volume (1 × 10$^8$ m$^3$) | Adjustment Parameters of CN Values | Time (Month) |
|---|---|---|---|---|---|---|---|---|
| Calibration (R$^2$ = 0.95, NSE = 0.94) | Guchengzi | 124.260 | 48.533 | 25,485 | 6.98 | 7.01 | 0.88 | 1–12 |
| | Changjiangtun | 129.592 | 45.990 | 35,465 | 32.00 | 33.15 | 0.97 | 1–12 |
| | Linghai | 121.367 | 41.183 | 22,286 | 0.57 | 0.71 | 0.70 | 5–9 |
| | Yanchi | 115.883 | 40.033 | 52,094 | 0.19 | 0.27 | 0.36 | 6–9 |
| | Jingcun | 108.137 | 35.013 | 40,333 | 2.34 | 3.20 | 0.47 | 1–12 |
| | Geermu | 94.780 | 36.307 | 20,042 | 1.66 | 1.91 | 0.47 | 6–9 |
| | Chaoyang | 101.760 | 36.657 | 38,205 | 7.04 | 7.13 | 0.57 | 1–12 |
| | Feilaixia | 113.236 | 23.786 | 36,899 | 76.80 | 76.05 | 1.08 | 1–12 |
| | Wujingdu | 106.787 | 27.314 | 24,643 | 55.10 | 55.08 | 1.11 | 1–12 |
| | Geputan | 113.717 | 30.938 | 8730 | 13.40 | 13.22 | 1.11 | 1–12 |
| | Xiaolongtan | 103.186 | 23.814 | 187,867 | 9.17 | 9.53 | 0.82 | 1–12 |
| | Jiajiang | 103.543 | 29.753 | 12,540 | 31.67 | 32.87 | 1.15 | 1–12 |
| | Tangjia | 91.793 | 29.899 | 20,046 | 19.04 | 19.03 | 0.95 | 5–10 |
| | Lijiadu | 116.161 | 28.215 | 15,855 | 50.32 | 50.67 | 1.16 | 1–12 |
| | Yangkou | 117.918 | 26.796 | 12,521 | 76.19 | 51.74 | 1.16 | 1–12 |
| Validation (R$^2$ = 0.96, NSE = 0.93) | Dashankou | 85.734 | 42.251 | 18,568 | 9.03 | 8.65 | 1.06 | 6–9 |
| | Heishiguan | 112.931 | 34.719 | 18,579 | 6.03 | 6 | 0.65 | 1–12 |
| | Kafuqihai | 82.484 | 43.422 | 19,067 | 29.80 | 37.78 | 0.99 | 5–8 |
| | Chenming | 129.483 | 46.973 | 20,272 | 12.31 | 9.97 | 0.92 | 4–11 |
| | Yitang | 111.833 | 37.001 | 23,876 | 2.15 | 1.53 | 0.45 | 6–9 |
| | Ganzi | 99.967 | 31.619 | 33,720 | 0.44 | 0.09 | 0.76 | 1–3 |
| | Shimen | 111.384 | 29.588 | 15,584 | 38.53 | 40.02 | 1.11 | 1–12 |
| | Lanxi | 119.468 | 29.218 | 12,789 | 30.26 | 27.13 | 1.08 | 1–12 |
| | Gengzhang | 94.152 | 29.746 | 15,030 | 9.84 | 6.89 | 0.93 | 5–10 |

The 18 watersheds were transformed into feature points to obtain geometric centers with CN-value adjustment parameters. The ordinary Kriging interpolation method was used to obtain adjustment parameters nationally, and CN-value adjustment parameters of each city were calculated by averaging parameter values within a city (Figure 6) [57]. CN values of each land use type in each watershed were then calculated by multiply the averaged adjustment parameters and the initial values. CN values greater than 98 after the adjustment were defined to be 98.

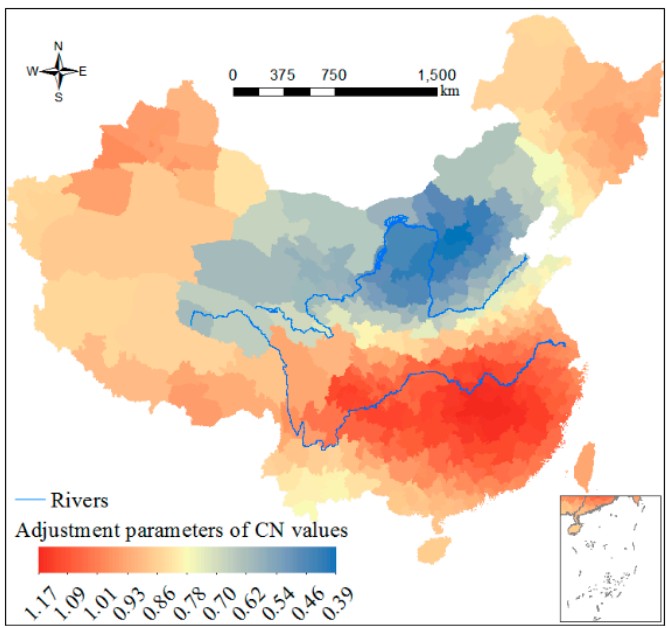

**Figure 6.** Adjustment parameters of CN values in each city.

(2) Model Validation

Similarly, the averaged adjustment parameters of CN values of 7 validation watersheds were calculated by applying the averaged adjustment parameters within each of the watershed (Table 3). CN values of each land use type in each watershed were then calculated by multiply the adjustment parameters by the initial values. Then, surface runoff volumes of the 7 watersheds in 2016 were simulated with the calibrated L-THIA model for validation. $R^2$ and NSE were calculated as 0.96 and 0.93, respectively. Values of $R^2 \geq 0.6$ and NSE $\geq 0.5$ are regarded as indicating good model performance [55]. Therefore, the model in this study after calibration was regarded as performing well.

2.2.5. Scenario Simulation and Assessment of LUCC and Rainfall Change on Surface Runoff

Three scenarios were modeled to study the impacts of LUCC and rainfall change on surface runoff. Being the major variables, configurations of LUCC and rainfall data in each scenario were as follows (Table 4).

(1) S1: Assessment of the responses of surface runoff depth to LUCC and rainfall change

Study of the impacts of LUCC and rainfall change on surface runoff depth is the initial motivation of this study. Average annual surface runoff volume of 2005, 2010 and 2015 were assessed with the land use data and rainfall data. Five-year interval rainfall data were used to reflect the variation trend of rainfall in 15 years from 2003 to 2017. Then, the average surface runoff depth of each city could be calculated with the following formula:

$$\mathrm{ARD_{city}} = (\mathrm{TARV_{city}} / A_{\mathrm{city}}) \times 1000 \tag{6}$$

$\text{ARD}_{\text{city}}$ (mm) is the annual runoff depth of each city, $\text{TARV}_{city}$ (m$^3$) is total annual runoff volume of each city, and $A_{\text{city}}$ (m$^2$) is the area of each city.

Annual runoff depths of each city in 2005, 2010 and 2015 were calculated to analyze the spatial-temporal change of surface runoff in China. Annual runoff depth change of each city from 2005 to 2010, and 2010 to 2015 were also calculated to analyze the change tendency in the last 10 years.

(2) S2: Simulation and assessment of the responses of surface runoff depth to rainfall change

Rainfall change occurs with strong randomness, and thus comparison between a single year's rainfall data cannot reflect rainfall change tendency. Annual surface runoff volumes of each city were first simulated with rainfall data from 2003 to 2017 and land use data of 2015. Then, the average values of annual runoff volumes in the first five years (2003–07), the second five years (2008–12) and the third five years (2013–17) were calculated separately. Increased annual runoff depths from 2005 to 2010 and 2010 to 2015 of each city were calculated for analyzing the impact of rainfall change on surface runoff.

(3) S3: Simulation and assessment of the responses of surface runoff depth to developed land expansion

One of the primary characteristics of urbanization is the increases in developed land area [58,59]. Transformation from grass land, woodland, and other natural land to developed land changes the physical characteristics of the surface greatly. Large areas of natural surface were changed into impervious surface, which adds to surface impermeability, and even the water permeability of green infrastructure land in urban areas decreases due to construction activities [60]. Influenced by human activities, land use/cover types changed greatly in recent decades [61,62]. Based on the land use dataset in this study, 54,072 km$^2$ (0.57% of the total area of China) of natural land were transformed into developed land from 2005 to 2015 nationally. To understand the impact of human construction activities on surface runoff, the annual runoff depth of the changed land before and after the transformation were simulated.

**Table 4.** Configuration of LUCC and rainfall data in each scenario.

| Scenarios | Input Data | Purpose |
|---|---|---|
| S1 | (1) LUCC data of 2005, rainfall data 2003–07 (average value of annual surface runoff volumes 2003–07 was used); (2) LUCC data of 2010, rainfall data 2008–12 (average value of annual surface runoff volumes 2008–12 was used); (3) LUCC data of 2015, rainfall data 2013–17 (average value of annual surface runoff volume 2013–17 was used); | To assess the responses of surface runoff depth to LUCC and rainfall change. |
| S2 | (1) LUCC data of 2015, rainfall data 2003–07 (average value of annual surface runoff volumes 2003–07 was used); (2) LUCC data of 2015, rainfall data 2008–12 (average value of annual surface runoff volumes 2008–12 was used); (3) LUCC data of 2015, rainfall data 2013–17 (average value of annual surface runoff volume 2013–17 was used); | To assess the impacts of rainfall change on surface runoff depth. |
| S3 | (1) The increased developed land 2005–10, rainfall data of 2017; (2) The increased developed land 2010–15, rainfall data of 2017; (3) The initial land types of 2005 that transformed into developed land in 2010, rainfall data of 2017; (4) The initial land types of 2010 that transformed into developed land in 2015, rainfall data of 2017; | To assess the impacts of developed land expansion on surface runoff depth. |

## 3. Results and Discussion

### 3.1. Scenario Simulation Results

(1) S1: Assessment results of the response of surface runoff depth to LUCC and rainfall change

Figure 7 shows the change tendency of annual surface runoff depth in each city from 2005 to 2015 with LUCC and rainfall change. Influenced by the LUCC and rainfall change in the last 15 years,

the surface runoff did not show obvious geographical changes. Generally, high values of runoff depths were located in the middle and lower reaches of the Yangtze River, low values of runoff depths were located in the Yellow River basin and southwest China, and the middle values of runoff depths were located in northwest, northeast and the south coast of China. However, the average annual runoff depth increased nationally with values of 78 mm, 83 mm and 90 mm in 2005, 2010 and 2015 respectively. For example (Figure 8a), in 2005, 110 cities had increased annual runoff depths greater than 100 mm, but this increased to 114 cities and 120 cities in 2010 and 2015, respectively. Fifteen cities had increased annual runoff depths greater than 600 mm in 2005, with an increase to 24 cities and 26 cities in 2010 and 2015.

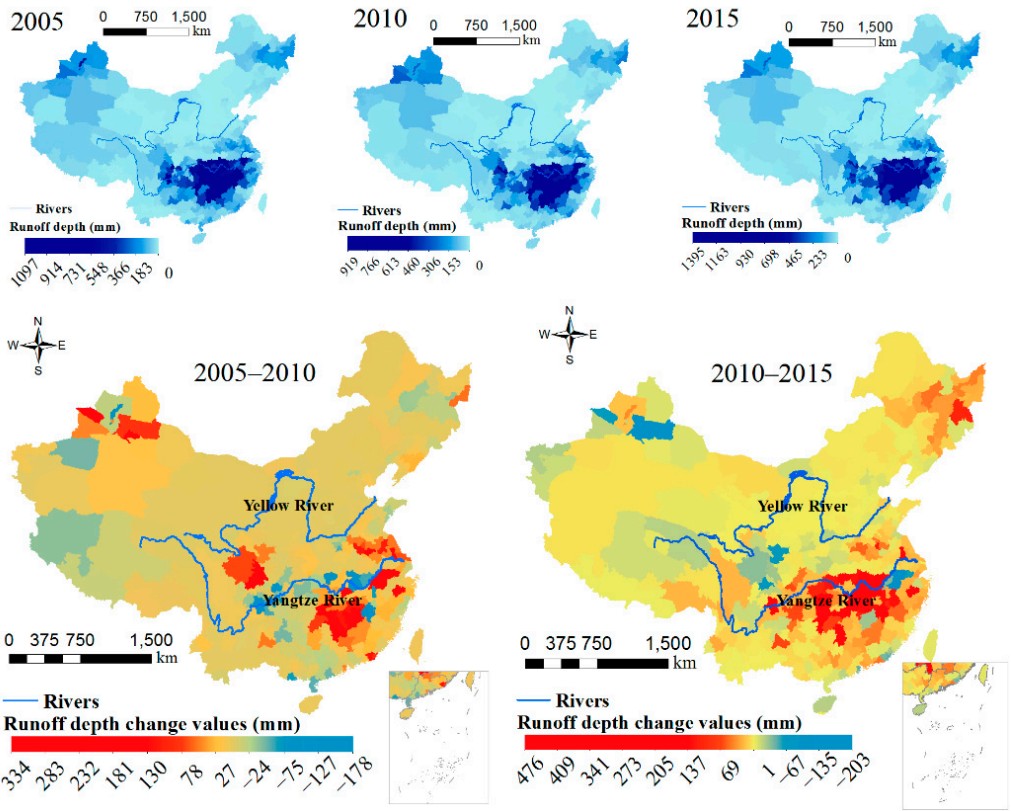

**Figure 7.** S1: Change of annual runoff depth in each city with LUCC and rainfall change 2005–2010, 2010–2015 and 2005–2015.

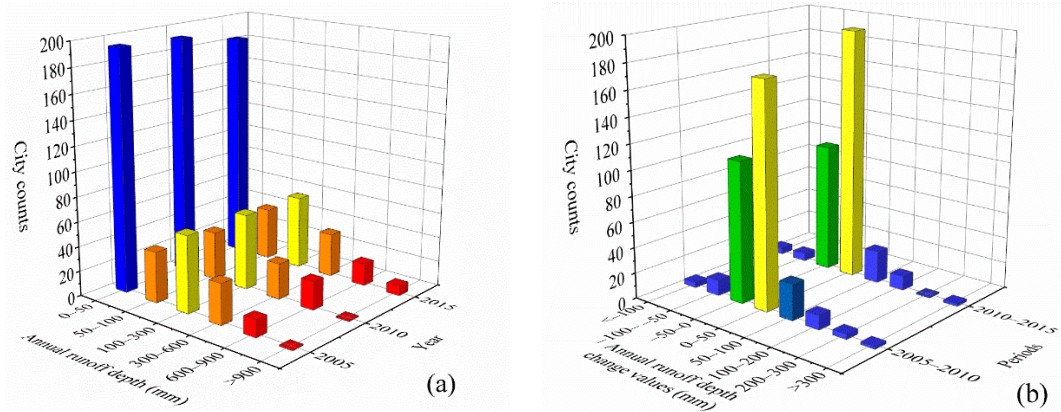

**Figure 8.** S1: (**a**) Annual runoff depth statistics in 2005, 2010 and 2015; S1: (**b**) annual runoff depth change value statistics 2005–10 and 2010–15.

Figure 7 also shows the annual runoff depth change values from 2005 to 2010 and 2010 to 2015. Just like the distribution of annual runoff depths by geography, high values of annual runoff depth change were mainly located in the middle and lower reaches of the Yangtze River. Some cities located in northwest China and the upper reaches of the Yellow River also had high vales from 2005 to 2010. Some cities located in northeast China had high values from 2010 to 2015. Based on the statistical results in Figure 8b, there were 219 cities that had increased annual runoff depths, of which 16 cities had values greater than 100 mm in the first period. In the second period, there were 234 cities that had increased annual runoff depths, of which 14 cities had values greater than 100 mm.

(2) S2: Response of surface runoff depth to rainfall change

Figure 9 shows that when the land use data are fixed, different rainfall data can cause large changes in surface runoff depth. From 2005 to 2010, annual runoff depth change could be as large as 331 mm, while this value was up to 476 mm from 2010 to 2015. As the statistics show (Figure 10a), in the first period, there were 45 cities which had increased annual runoff depths greater than 50 mm, and 17 cities which had decreased annual runoff depths greater than 50 mm. In the second period, city counts with increased and decreased annual runoff depths greater than 50 mm were 38 and 12, respectively.

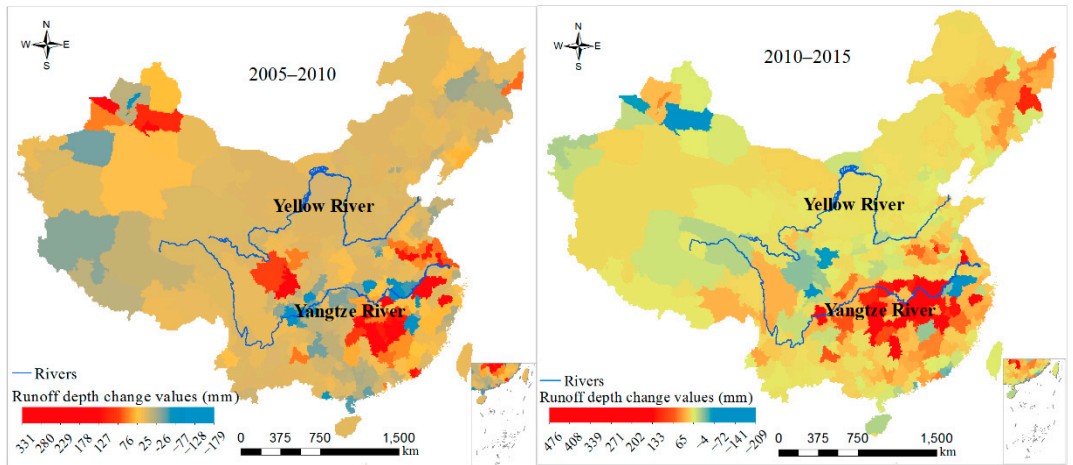

**Figure 9.** S2: Annual surface runoff depth change with rainfall change from 2005 to 2010 and 2010 to 2015.

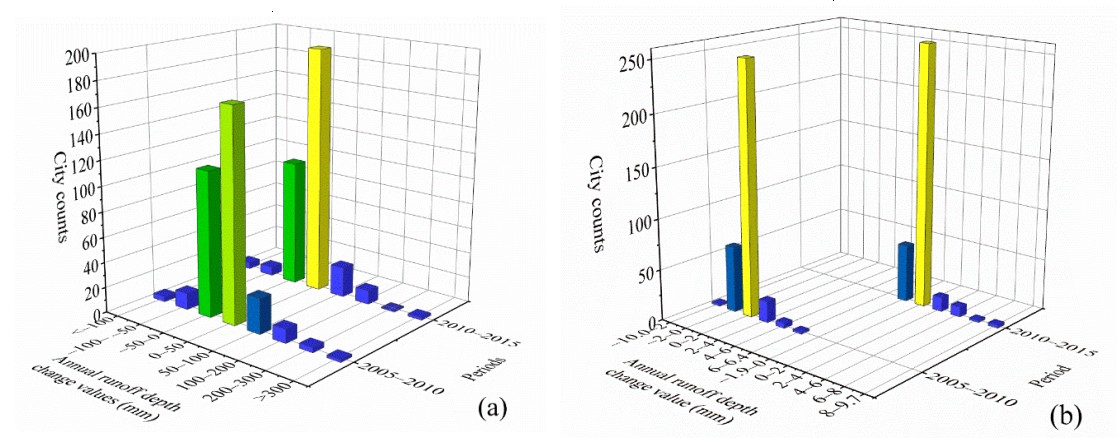

**Figure 10.** S2: (**a**) Annual runoff depth change values from 2005 to 2010 and 2010 to 2015 with rainfall change; S2−S1: (**b**) Difference of the results after and before taking land use as the variable.

Compared with the simulation results of the response of surface runoff depth to LUCC and rainfall change (Figure 8b), we found that the LUCC may or may not be the variate, as the simulation

results did not show much difference. To quantify the difference, subtraction was applied with the annual runoff depth change values of each city in the two periods with LUCC being the variate minus the ones with LUCC not being the variate (Figure 10b). Results show that when LUCC was one of the variates to simulate surface runoff depth, the annual runoff depth change values would increase by −10 mm to 6.4 mm from 2005 to 2010, of which 315 cities would increase by −2 mm to 2.0 mm. Correspondingly, the annual runoff depth change value would increase by −1.9 mm to 9.7 mm from 2010 to 2015, and 316 cites would increase by −1.9 mm to 2.0 mm. Nationally, with the LUCC being the variate, the average annual runoff depth change value increased by 0.47 mm and 0.57 mm in two periods. Based on this analysis, LUCC had less impact on surface runoff change than rainfall change.

Although the influence of LUCC on surface runoff change was small, land use types with different soil characteristics greatly influence surface runoff. For example, the annual rainfall depths of the northwest and northeast China were small (Figure 1), but the average annual runoff depths and the annual runoff depth change values in the two periods were relatively high (Figure 9).

(3) S3: Response of surface runoff depth to developed land expansion

China is experiencing rapid urbanization in recent decades [58]. Based on the dataset in this study, the total developed land area increased by 26,552.7 km$^2$ and 27,519.1km$^2$ from 2005 to 2010 and 2010 to 2015, respectively. However, the increased developed land was mainly located in eastern China, especially the lower reaches of the Yellow River and Yangtze River (Figure 11). Some cities even had decreased developed land area (every city experienced developed land change, only increased developed land were used for surface runoff simulation in this study). Human construction activities changed the characteristics of the natural land, but the impact on surface runoff differed geographically (Figure 11). Results show the influence of rainfall, characteristics of the soil, and developed land expansion of the middle and lower reaches of the Yangtze River caused great change of annual runoff depth, followed by the northeast and northwest China. On the contrary, developed land expansion of other parts did not cause obvious annual runoff depth change.

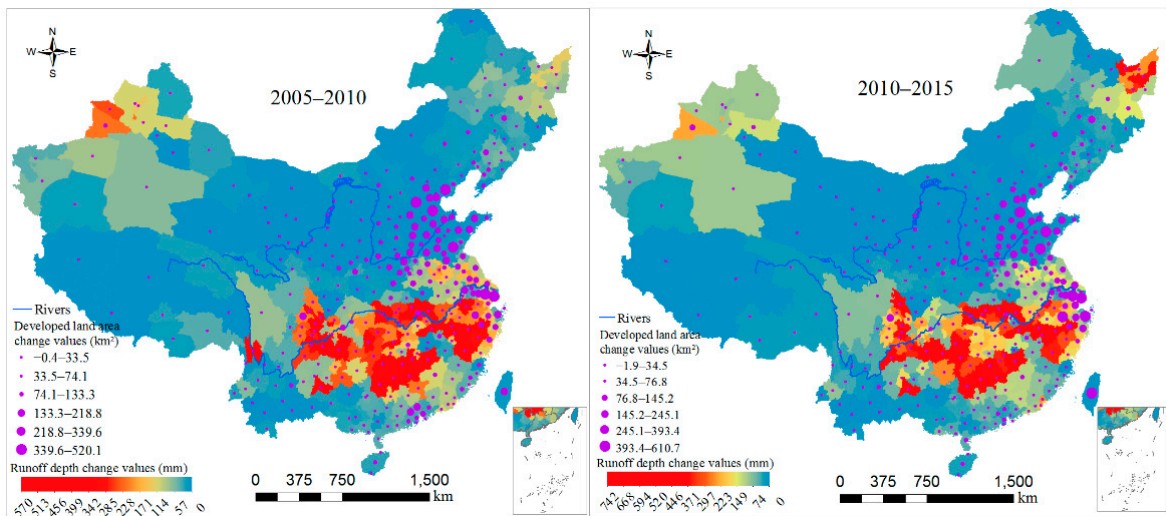

**Figure 11.** S3: Annual surface runoff depth change with developed land expansion from 2005 to 2010 and 2010 to 2015.

Developed land expansion of the two periods also impacted annual runoff depth change values (Figure 12). Nationally, developed land expansion from 2005 to 2010 and 2010 to 2015 caused annual average runoff depth increases by 88 mm and 96 mm compared with the initial natural land. Specially, there were 93 and 99 cities in these two periods that had increased runoff depths greater than 100 mm, and 33 and 37 cities had increased annual runoff depths greater than 300 mm. In addition,

the maximum values of the increased annual runoff depth also differed; the one for the first period was 570 mm and the one of the second period was 742 mm.

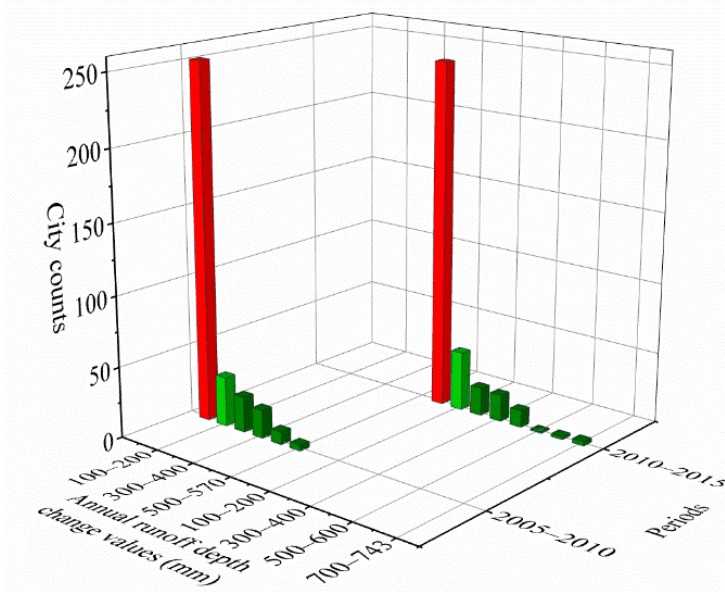

**Figure 12.** S3: Annual surface runoff depth change statistics with LUCC from 2005 to 2010 and 2010 to 2015.

### 3.2. Thoughts About Urban Land Use and Surface Runoff Control with Urbanization

Rainfall change and LUCC have different effects on surface runoff. In comparison, the spatial-temporal change of rainfall has a greater impact on regional average surface runoff, while the impact of LUCC on regional surface runoff is relatively small. However, on the micro scale, developed land expansion can greatly change the original surface characteristics and runoff, thus affecting the ecological environment and human activities. Therefore, control and management of surface runoff have become a major problem for urban land expansion. Surface runoff control and management concepts such as low-impact development practices, sponge city construction, and eco-city construction are widely used in urban areas [63,64]. Based on the results of this study, several suggestions about urban flood management were proposed: (1) Concepts of flood management in different areas should be different. Areas with less rainfall and that do not generate significant surface runoff should focus on rainwater collection and reuse [65], while areas with more rainfall that easily generate surface runoff should pay more attention to the rapid collection and discharge of rainwater. Areas with less rainwater and that easily generate surface runoff should focus on the rapid collection of rainwater and the reuse of rainwater simultaneously. (2) Multiple flood management objectives should be encouraged [66]. Objectives to reuse or collect rainwater in single rain events should be different in areas with great variability in rainfall. Additionally, areas with similar rainfall but different hydrologic soil group types should also be various in objectives. (3) Practices to treat urban floods should be different. Practices addressing urban floods have different impacts on water collection, retention and infiltration. Practice selection should be based on the flood management objectives [65,67]. (4) Integration of multiple planning approaches is needed, such as land use planning, urban construction planning, and flood management planning [68]. Linkage of multiple planning efforts can reduce redundant construction, avoid conflicting planning objectives, and achieve effective land use and flood management.

## 4. Conclusions

China is experiencing rapid urbanization and developed land expansion, which are important driving forces changing surface runoff. The SCS-CN method and L-THIA model were used to assess the impact of LUCC and rainfall change on surface runoff depth from 2005 to 2015 in China. Conclusions are as follows: (1) Distribution of annual runoff depth by geographic region from 2005 to 2015 did not show obvious changes, with high values mainly located in the middle and lower reaches of the Yangtze River. However, the average runoff depths nationwide in 2005, 2010 and 2015 increased from 78 mm to 83 mm to 90 mm. (2) Changes of annual runoff depth nationally were mainly caused by rainfall change within the research period of the study, and LUCC played a relatively less important role in annual runoff depth change. Generally, with the rainfall data of 2017 fixed, LUCC from 2005 to 2010 and 2010 to 2015 caused changes in annual runoff depths by 0.47 mm and 0.57 mm, respectively. (3) Nationally, developed land expansion in different regions from 2005 to 2010 and 2010 to 2015 led to annual runoff depth change ranging from 0 mm to 570 mm and 0 mm to 742 mm, respectively. (4) Urban flood management concept, objectives and practices were encouraged to be different to treat urban surface runoff in different regions. Integration of multiple planning was also needed to treat the contradiction of LUCC and surface runoff change.

This study also demonstrated an approach to reclassifying soil texture and adjusting the CN values based on adjustment parameters nationally to accomplish the application of the L-THIA model in China. CN adjustment was based on average values of cities, while the locations of cities varies greatly, which limited the accuracy of simulation results. Further study could be focused on improving the CN value with higher resolution of adjustment units. In addition, the HSG dataset is also an important input data which influences the CN values greatly; more study can also focus on using higher resolution of soil dataset. Short-term rainfall data in this study could not represent long-term climate variability, so a stochastic weather generator could be used to obtain continuous long-term rainfall time series data to study the impact of long-term climate variability on surface runoff. Future land use change scenarios and their hydrological and environmental impacts could also be carried out, based on simulated land use data and rainfall data.

**Author Contributions:** All authors contributed to the design and development of this manuscript. Conceptualization and writing (review and editing), J.C.; Formal analysis, H.S.; Methodology, Y.L.; Resources, S.W. and P.X.; Writing—original draft, F.L.; Writing—review & editing, B.A.E.

**Funding:** This work is supported by the Overseas Expertise Introduction Project for Discipline Innovation (111 project) (Grant No. B17024), Research Fund Program of Guangdong Provincial Key Laboratory of Environmental Pollution Control and Remediation Technology (Grant No. 2018K01), the Postgraduate Research & Practice Innovation Program of Jiangsu Province (Grant No. KYCX17_0641).

**Acknowledgments:** Thanks to the European Space Agency (ESA) Climate Change Initiative (CCI)—Land Cover project for providing the land use dataset.

**Conflicts of Interest:** The authors declare no conflicts of interest.

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
