# Peer review of "Assessment of the Impacts of Land Use/Cover Change and Rainfall Change on Surface Runoff in China"

_sustainability, doi:10.3390/su11133535_

Round 1

Reviewer 1 Report

This manuscript deals with a study utilizing L-THIA to evaluate the effects of land use change and rainfall pattern change on surface runoff. The reviewer has the following comments:

1. Line 20: It is more appropriate to say “…understanding of the temporal trend of variation…”

2. In the abstract, the reviewer suggests also highlighting that a conversion system of soil classification is proposed in the manuscript.

3. Fig.1: For the developed land area, the reviewer suggests using “percentage developed” instead because the total land area of each province is drastically different.

4. Lines 164-176: Please state the purpose of the cell numbering in the beginning of this paragraph. It appears that the numbering system does not have a functional purpose except for internal tagging for cells.

5. Table 3: Please provide reference for the equations used in Table 3, as the reviewer cannot find these equations in [45] or [46]. Please advise.

6. Line 215: The calibration accuracy is astonishingly high. According to earlier descriptions, it is based on daily rainfall. Please confirm this is right. Please provide more detailed calibration results such as a graph or table. The same request goes to the validation as well.

7. Table 5: Please put horizontal separator between S1, S2, and S3 scenarios.

8. Table 5: The reviewer does not understand the purpose of S1 scenarios. Why do you change both land use and rainfall among the scenarios? How does it tell you the influence of land use or rainfall alone?

9. Table 5: There are two Table 5 (lines 206 vs. 227)

10. Font size in figures are sometimes too small. Please make sure that all text in figures are no smaller than that in Fig. 4.

11. Figures 7-12: Please label the number of scenarios they belong to. Also, the viewing angle in Figures 8, 10, and 12 made some words not easy to recognize. Please find a way to improve it.

12. Figure 8: (a) and (b) in the caption are at the wrong locations.

13. The reviewer suggests including some of “Thoughts About Urban Land Use and Surface Runoff Control with Urbanization” into the conclusion.

Author Response

1. Line 20: It is more appropriate to say “…understanding of the temporal trend of variation…”

Response: Improved

2. In the abstract, the reviewer suggests also highlighting that a conversion system of soil classification is proposed in the manuscript.

Response: Improved

3. Fig.1: For the developed land area, the reviewer suggests using “percentage developed” instead because the total land area of each province is drastically different.

Response: Improved

4. Lines 164-176: Please state the purpose of the cell numbering in the beginning of this paragraph. It appears that the numbering system does not have a functional purpose except for internal tagging for cells.

Response: “Pixels with same attributes were distribute to be one hydrology unit in this tool, while it’s incapable in calculating the runoff volumes of each assessment unit. To solve this problem, a numbering system was used to improve the tool.”

5. Table 3: Please provide reference for the equations used in Table 3, as the reviewer cannot find these equations in [45] or [46]. Please advise.

Response: The equations in Table 3 can not be find in citations, they were defined according to the description of each land use type. CN values of each land use types were first defined based on the relationships between these two classification systems. Of which, the CN values of pure land use types (with single kind of land use type in each pixel) were defined to be same with the US land use/cover types. CN values of other land use types that with mixed types were defined by the weighted values.

6. Line 215: The calibration accuracy is astonishingly high. According to earlier descriptions, it is based on daily rainfall. Please confirm this is right. Please provide more detailed calibration results such as a graph or table. The same request goes to the validation as well.

Response: We understand the doubt of the reviewer about the high accuracy. The R2 and NSE were calculated based on the summary values of single year’s or multiple months’ surface volume of each watershed. All the simulated surface runoff volumes of the 17 watersheds (for model calibration) were all calibrated based the observed values one by one (adjust the CN values 1% each time), until the simulated yearly values matched the observed yearly values. That why the calibration accuracy is high.

We also compared the daily simulated results with the observed surface volume, however we found that, the error of each watershed is relatively large. We think that’s may be because the observed daily values were also simulated results by applying the BFLOW program. Comparison between two sets of simulated daily values cause big errors, but when the time span is long enough, the relatively error of the total values would become smaller. Considering that the purpose of this study was to evaluate the annual impact LUCC and rainfall change, and all the results in this study were measured by annual values. We did not discuss much on the error of daily results. Similar research methods could also be found in the study of Liu, et al. [1].

[1].   Liu, Y.; Bralts, V. F.; Engel, B. A., Evaluating the effectiveness of management practices on hydrology and water quality at watershed scale with a rainfall-runoff model. Sci Total Environ 2015, 511, 298-308. Doi.org/10.1016/j.scitotenv.2014.12.077

As for the model validation, annual values of the 8 watersheds were also used for calculated the R2 and NSE.

7. Table 5: Please put horizontal separator between S1, S2, and S3 scenarios.

Response: Improved

8. Table 5: The reviewer does not understand the purpose of S1 scenarios. Why do you change both land use and rainfall among the scenarios? How does it tell you the influence of land use or rainfall alone?

Response: Considering that rainfall change in different year may also occurs with randomness, five-year average values of surface runoff volumes were used to reflect the variation trend of rainfall in the 15 years.

Influence of LUCC and rainfall change alone were processed in S3 and S2 respectively.

9. Table 5: There are two Table 5 (lines 206 vs. 227)

Response: improved

10. Font size in figures are sometimes too small. Please make sure that all text in figures are no smaller than that in Fig. 4.

Response: improved

11. Figures 7-12: Please label the number of scenarios they belong to. Also, the viewing angle in Figures 8, 10, and 12 made some words not easy to recognize. Please find a way to improve it.

Response: Scenario numbers were labeled in each title of the figures.

All the figures were improved to make the words easy to recognize.

12. Figure 8: (a) and (b) in the caption are at the wrong locations.

Response: Improved

13. The reviewer suggests including some of “Thoughts About Urban Land Use and Surface Runoff Control with Urbanization” into the conclusion.

Response: Improved

Reviewer 2 Report

The manuscript was extensively revised and it is now substantially improved. Many justifications and clarifications were added and the calibration and validation parts are now very good and a great enhancement of the overall study quality. I only have a few remaining comments that I include in the commented pdf file. In case that the authors address these remaining minor comments, I believe that it can be published. Please also make a final check of the language.

Author Response

Response: Thanks so much for the comments. Some main comments in the pdf file were listed as follows:

1. Line 151: add some citation to papers discussing the characteristics and the pros and cons of SCS-CN method.

Response: Added

2. line194: I can understand that there is a reason behind this. You should provide the reasons /justification.

Response: Detailed explanation was added: “Specially, surface runoff generated by perennial ice/snow were not included in this study and CN values perennial ice/snow were then defined to be zero. For, perennial ice/snow is mainly located on northwest area with high altitude, low temperature, relatively small rainfall depth and small area (0.7% of the total area). Surface runoff generation process of perennial ice/snow is also different from other land use types while relative research is rare. In addition, the purpose of this study is to evaluate surface runoff of the ground, and the observed surface runoff volumes used for model calibration and verification in following study were also estimated surface runoff of the ground by applying the Baseflow Filter Program (BFLOW) [51]. Thus, CN values of water surface was also defined to be zero in this study.”

3.line 199: Add one or more citations

Response: added.

4. line 318: You may discuss also that runoff increments due to LUCC can be more important in specific locations (small scale) urbanized areas even if the average change (large scale) is low. This may result in flood risk increment in specific places.

Response: the impact of developed land expansion on surface runoff volume was discussed in detail in S3: line341 to line359.

5. line 371: It would be great if you could add some citations for some of these suggestions.

Response: added. 

Reviewer 3 Report

Dear authors,

I´m glad about the changes made from the first version to this one.

Therefore, I recommend the acceptance of the manuscript 

best,

Author Response

Dear reviewer,

Thanks so much for your all work for this paper.

Best wishes.

This manuscript is a resubmission of an earlier submission. The following is a list of the peer review reports and author responses from that submission.

Round 1

Reviewer 1 Report

This manuscript discussed a nationwide investigation of the impact of rainfall variation and land use change on runoff potential using the CN method in L-THIA. The topic of this manuscript has good academic value, but the methodology and writing suffer from two major issues:

The first main issue of this study is the lack of validation of the results. Since some variation of soil classification and significant manipulation of land use categorization was involved in modeling, some kind of validation of the result is necessary. I would recommend using small-scale observation data, whichever available, to validate the modeling accuracy before applying it to the national scale.

The second issue of this study is the chaotic presentation of scenarios. There are two major variables: LUCC and rainfall. Multiple scenarios were modeled to study the effect of each variable. However, the authors failed to first present a comprehensive summary in how scenarios were developed to investigate these two variables in the methodology section. The author directly presented the results of scenarios in the results section without any introduction, which makes it difficult to read. The authors also failed to clearly provide configurations of each scenario in text, so it is difficult to understand how these two variables are controlled (i.e. fixed or varied) in each scenario.

Other issues, along with the two major ones discussed above, are listed below:

Lines 25-29: Please rearrange the focus of assessment using simple cause-effect relationships (e.g. “…to evaluate the variation of X, Y, and Z, and further evaluate the effect of A on B, C on D, and E on F”). If possible, use separate sentences to describe goals of different nature. The current way of description is confusing and makes the reader hard to comprehend the main focus of this manuscript.

Line 49: Grammatically, “waterlogging disasters” is not wrong, but it is seldom used this way. Please simply use “floods”.

Line 50: I would suggest adding the loss in equivalent U.S. dollars so readers in other countries can understand the magnitude.

Line 64: The cited references do not contain one that involves SWMM, but SWMM was mentioned in the text. I would suggest including the following paper here:

Tu, M.-c.; Smith, P. 2018. Modelling pollutant buildup and washoff parameters for SWMM based on land use in a semiarid urban watershed. Water Air Soil Poll. 229: 121. doi:10.1007/s11270-018-3777-2                

Lines 85-92: I would recommend having the sources cited in the references section.

Line 109: The use of “overlay” is ambiguous here in terms of what actions/calculations were actually taken. Please consider revising.

Lines 127, 133: I would suggest using another word to describe “F”. People often use infiltration or soil absorption to describe F. Retention can be misunderstood as water retention by stormwater structures.

Line 135: Ia=XS where X=0.2 may be reasonable in the natural environment, but it is too large for the highly urbanized urban region in China. Please check whether that value in L-THIA can be adjusted, and use a variable value for X. I think TR-55 published a relationship between Ia and CN. If this study was for a single localized area, the use of a single X might be OK, but this study encompasses the whole country so ignoring the difference in initial abstraction can distort the result.

Line 141: The term “pixel” usually refers to the minimum unit of image displaying, which might cause confusion in this study where graphical illustrations are used intensively. In this case, “cell” is a more suitable word. Please revise this term for the whole manuscript.

Lines 151-161: This is confusing. Please consider using a new graph to illustrate the process. If this is not essential in modeling process or results, please consider deleting it.

Lines 169-171: This is problematic. To study the influence of a single factor (e.g. LUCC), other factors should be fixed. However, you are varying both rainfall and LUCC in your modeling.

At later sections, it appears that you have scenarios where rainfall data is fixed, but it was not described clearly in the methodology. As discussed above, I strongly recommend creating a new section in the methodology to describe the design of scenarios and reasons to have these scenarios.

Also, rainfall data of merely 15 years is not enough to investigate the effect of climate change. If there is any change, it is more likely to be short-term. I would suggest framing the rainfall change in this study as the short-term decadal rainfall pattern trend, instead of climate change. Can you plot the rainfall data in the manuscript so we can see if there is any pattern?

On the same vein, you need to plot the temporal change of developed land from 2005-2015 so readers can see the actual fluctuation of the influencer variable of LUCC. You have it in Fig 5, but you need to mention it before presenting any results, as it is your input data.

Line 187: I recommend using “developed land” to replace “construction land” in the manuscript.  The term “construction land” usually means the land under construction.

Line 198: At line 192, ARVGCL is runoff volume, but it becomes area here?

Line 199: “…was defined to present the…”

Line 212: I might have missed it, but where did you mention this constant rainfall scenario?

Figure 3: 1) You need a colon to separate “A, B, and C” (also “D, E, and F”) and the corresponding description. 2) “…change of annual surface runoff volume with rainfall change from 2005 to 2010, 2010 to 2015, and 2005 to 2015, respectively”. Please change similar parts accordingly. 3) If my understanding is right, A, B, and C display scenarios where LUCC is changed but rainfall is fixed, while C, D, and E display scenarios that LUCC is fixed and rainfall is changed? It is not clear in either the caption or corresponding text. Like mentioned before, if you clearly define and summarize the configurations of each scenario in the methodology section, there would be much less confusion here.

Lines 237-240: 1) How Figure 4 was made is not clearly expressed. Is Fig 4(A) = Fig 3(A) + FIG 3(D)? 2) How did you derive the conclusion that “LUCC data had little impact on the surface runoff distribution, and rainfall data had a decisive impact”? I see that D-F have more red, but you need to guide readers through your thought process. Please provide detailed reasoning.

Lines 281-284: Please break the part after the semicolon to a new sentence.

Line 282: “…in the second period, of which 16 provinces (Anhui, …, and Yunnan) showed a sustained increase in both periods.”

Section 3.4: After the results in 3.1-3.3, some statistical analysis is in order. In addition to displaying changes in runoff and development land for each city and province (like you did in 3.1-3.3), you should develop a few scatter plots to show the relation between rainfall pattern, developed area, and runoff volume across the whole nation and/or regions.

Line 322: “…spatially with higher development speed and quality in the east than in the west.”

Line 323: “…increasing rates of…are higher…”

Lines 333-341: In addition to what you mentioned, I believe the initiative of “sponge city” also addressed a phenomenon you found earlier. You found that development land area is not the major influencer to runoff volume. I believe it can be explained, at least partly, by the adoption of LIDs in “sponge cities”. Please add this in an appropriate location in the manuscript.

Line 348: Please break the part of the sentence after the semicolon into a new one. I have to say that the authors like to use semicolons to create super long sentences. This is not a good writing practice. Concise and clarity are more important. Please search the whole manuscript and eliminate such use of semicolons.

Lines 351-357: It is intriguing that LUCC caused a more spatially widespread increase in runoff, but rainfall pattern change caused more increase in total runoff volume. Can you provide any explanation?

Lines 369-370: I don’t understand the link between low P value and the importance of the developed land area. P value is the portion of runoff created by development land. If it is low or even negative, it should indicate the opposite (i.e. developed area is NOT important)? Please develop more reasoning in the manuscript.

Lines 371-372: So what does it tell us?

Author Response

Response to Reviewer 1 Comments

(1) The first main issue of this study is the lack of validation of the results. Since some variation of soil classification and significant manipulation of land use categorization was involved in modeling, some kind of validation of the result is necessary. I would recommend using small-scale observation data, whichever available, to validate the modeling accuracy before applying it to the national scale.

Response 1: Two small watershed were used to verify the model. (section 3.1)

(2) The second issue of this study is the chaotic presentation of scenarios. There are two major variables: LUCC and rainfall. Multiple scenarios were modeled to study the effect of each variable. However, the authors failed to first present a comprehensive summary in how scenarios were developed to investigate these two variables in the methodology section. The author directly presented the results of scenarios in the results section without any introduction, which makes it difficult to read. The authors also failed to clearly provide configurations of each scenario in text, so it is difficult to understand how these two variables are controlled (i.e. fixed or varied) in each scenario.

Response 2: I added a part of content to make it clear (Table4)

(3) Lines 25-29: Please rearrange the focus of assessment using simple cause-effect relationships (e.g. “…to evaluate the variation of X, Y, and Z, and further evaluate the effect of A on B, C on D, and E on F”). If possible, use separate sentences to describe goals of different nature. The current way of description is confusing and makes the reader hard to comprehend the main focus of this manuscript.

Response 3: Improved

(4) Line 49: Grammatically, “waterlogging disasters” is not wrong, but it is seldom used this way. Please simply use “floods”.

Response 4: Improved

(5) Line 50: I would suggest adding the loss in equivalent U.S. dollars so readers in other countries can understand the magnitude.

Response 5: Improved

(6) Line 64: The cited references do not contain one that involves SWMM, but SWMM was mentioned in the text. I would suggest including the following paper here:

Tu, M.-c.; Smith, P. 2018. Modelling pollutant buildup and washoff parameters for SWMM based on land use in a semiarid urban watershed. Water Air Soil Poll. 229: 121. doi:10.1007/s11270-018-3777-2

Response 6: Improved

 (7) Lines 85-92: I would recommend having the sources cited in the references section.

Response 7: The dataset used in this study can be downloaded directly following the website in the manuscript, so I didn’t change it just for simplify.

(8) Line 109: The use of “overlay” is ambiguous here in terms of what actions/calculations were actually taken. Please consider revising.

Response 8: Improved

 (9) Lines 127, 133: I would suggest using another word to describe “F”. People often use infiltration or soil absorption to describe F. Retention can be misunderstood as water retention by stormwater structures.

Response 9: I checked the original reference (The Technical Release 55) and found that, the word “retention” has been always used to describe “F”, and so, I still used the “retention” here.

(10) Line 135: Ia=XS where X=0.2 may be reasonable in the natural environment, but it is too large for the highly urbanized urban region in China. Please check whether that value in L-THIA can be adjusted, and use a variable value for X. I think TR-55 published a relationship between Ia and CN. If this study was for a single localized area, the use of a single X might be OK, but this study encompasses the whole country so ignoring the difference in initial abstraction can distort the result.

Response 10: The value of X is a default in the model, which can’t be adjusted. But I verified the model with two small watersheds.

(11) Line 141: The term “pixel” usually refers to the minimum unit of image displaying, which might cause confusion in this study where graphical illustrations are used intensively. In this case, “cell” is a more suitable word. Please revise this term for the whole manuscript.

Response 11: Improved

(12) Lines 151-161: This is confusing. Please consider using a new graph to illustrate the process. If this is not essential in modeling process or results, please consider deleting it.

Response 12: Improved. “Fig.4” was added to illustrate the process.

(13) Lines 169-171: This is problematic. To study the influence of a single factor (e.g. LUCC), other factors should be fixed. However, you are varying both rainfall and LUCC in your modeling.

At later sections, it appears that you have scenarios where rainfall data is fixed, but it was not described clearly in the methodology. As discussed above, I strongly recommend creating a new section in the methodology to describe the design of scenarios and reasons to have these scenarios.

Also, rainfall data of merely 15 years is not enough to investigate the effect of climate change. If there is any change, it is more likely to be short-term. I would suggest framing the rainfall change in this study as the short-term decadal rainfall pattern trend, instead of climate change. Can you plot the rainfall data in the manuscript so we can see if there is any pattern?

On the same vein, you need to plot the temporal change of developed land from 2005-2015 so readers can see the actual fluctuation of the influencer variable of LUCC. You have it in Fig 5, but you need to mention it before presenting any results, as it is your input data.

Response 13: Table.4 shows the configurations of each scenario. Rainfall change occurs with strong randomness, and thus comparison between a single year’s rainfall data cannot reflect rainfall change tendency. So, in scenario 2, annual surface runoff volumes of each city were first simulated with rainfall data from 2003 to 2017 and land use data for 2015. Then, the average values of annual runoff volumes in the first five years (from 2003 to 2007), the second five years (from 2008 to 2012) and the third five years (from 2013 to 2017) were calculated separately.

I add a Fig.1 in section 2.1 to describe the rainfall change and developed land area change.

(14) Line 187: I recommend using “developed land” to replace “construction land” in the manuscript.  The term “construction land” usually means the land under construction.

Response 14: Improved

(15) Line 198: At line 192, ARVGCL is runoff volume, but it becomes area here?

Response 15: Improved

(16) Line 199: “…was defined to present the…”

Response 16: Improved.

(17) Line 212: I might have missed it, but where did you mention this constant rainfall scenario?

Response 17: Improved. Table 4 was added in section 2.2.4 to describe the configuration of each scenario.

(18) Figure 3: 1) You need a colon to separate “A, B, and C” (also “D, E, and F”) and the corresponding description. 2) “…change of annual surface runoff volume with rainfall change from 2005 to 2010, 2010 to 2015, and 2005 to 2015, respectively”. Please change similar parts accordingly. 3) If my understanding is right, A, B, and C display scenarios where LUCC is changed but rainfall is fixed, while C, D, and E display scenarios that LUCC is fixed and rainfall is changed? It is not clear in either the caption or corresponding text. Like mentioned before, if you clearly define and summarize the configurations of each scenario in the methodology section, there would be much less confusion here.

Response 18: Improved. The former picture was divided into two little ones to make it clear.

(19) Lines 237-240: 1) How Figure 4 was made is not clearly expressed. Is Fig 4(A) = Fig 3(A) + FIG 3(D)? 2) How did you derive the conclusion that “LUCC data had little impact on the surface runoff distribution, and rainfall data had a decisive impact”? I see that D-F have more red, but you need to guide readers through your thought process. Please provide detailed reasoning.

Response 19: The previous statement was wrong. I have improved it here.

(20) Lines 281-284: Please break the part after the semicolon to a new sentence.

Response 20: Improved.

(21) Line 282: “…in the second period, of which 16 provinces (Anhui, …, and Yunnan) showed a sustained increase in both periods.”

Response 21: Improved.

(22) Section 3.4: After the results in 3.1-3.3, some statistical analysis is in order. In addition to displaying changes in runoff and development land for each city and province (like you did in 3.1-3.3), you should develop a few scatter plots to show the relation between rainfall pattern, developed area, and runoff volume across the whole nation and/or regions.

Response 22: I add a fig.10 to show the relation between rainfall pattern, developed area and runoff volume in section 3.4

(23) Line 322: “…spatially with higher development speed and quality in the east than in the west.”

Response 23: Improved.

(24) Line 323: “…increasing rates of…are higher…”

Response 24: Improved.

(25) Lines 333-341: In addition to what you mentioned, I believe the initiative of “sponge city” also addressed a phenomenon you found earlier. You found that development land area is not the major influencer to runoff volume. I believe it can be explained, at least partly, by the adoption of LIDs in “sponge cities”. Please add this in an appropriate location in the manuscript.

Response 25: The former statement “development land area is not the major influencer to runoff volume” is not accurate. I have improved the expression. Construction of “Sponge city” is just making progress in recent years, I don’t think it is the reason cause the decrease of runoff volume in some cities. But it is really a way to control surface runoff in urban areas.

(26) Line 348: Please break the part of the sentence after the semicolon into a new one. I have to say that the authors like to use semicolons to create super long sentences. This is not a good writing practice. Concise and clarity are more important. Please search the whole manuscript and eliminate such use of semicolons.

Response 26: Improved.

(27) Lines 351-357: It is intriguing that LUCC caused a more spatially widespread increase in runoff, but rainfall pattern change caused more increase in total runoff volume. Can you provide any explanation?

Response 27: I adjusted the CN values of the former model and re-simulated the runoff volume. Results changed a lot, I have improved the expression here.

(28) Lines 369-370: I don’t understand the link between low P value and the importance of the developed land area. P value is the portion of runoff created by development land. If it is low or even negative, it should indicate the opposite (i.e. developed area is NOT important)? Please develop more reasoning in the manuscript.

Response 28: Based on the re-simulated results, all the P values are greater than zero. The P value is the contribution rate of construction land change to annual runoff volume change in each province. This result may be easy to understand.

(29) Lines 371-372: So what does it tell us?

Response 29: improved

Reviewer 2 Report

In this study an assessment of the effect of land use change as well as precipitation change on runoff in a 10 years period (2005-2015) using the SCS-CN method in China is being presented. In general, the topic of the study is interesting and it is also interesting that the study investigates a very broad region dealing with China as a whole. The manuscript is well written and easy to understand. The paper seems also to be in-line with the scope and objectives of the journal.

On the other hand, the study has some important weaknesses narrowing its merit. The main weakness is that while the study is ambitious in terms of the size of the study area, the applied methodology is rather simplistic and does not allow for an in-depth assessment. SCS-CN method is powerful in assessing the effects of land use / land management changes, though, a more detailed and more elaborated application of the method is required in order to provide realistic results. For example, Hydrologic Soil Groups (HSG) were defined based on soil texture alone, while, more soil characteristics should be also considered such as soil depth. There is also no information in the study about how soil textural classes were corresponded with HSG and no justification in general. Similarly, there is a very coarse mapping between the available LU/LC categories of the Chinese dataset with the L-THIA model categories. In the documentation of SCS-CN method and in the numerous relevant studies there is plenty of information allowing a much better utilization of the available information in the Chinese LU/LC dataset (Table 2), allowing even the consideration of land management conditions. Here it should be noted that normally water bodies are assigned a CN value of 100 (impermeable areas), which is reasonable. In this study water bodies and wetlands were assigned a 0-value indicating no runoff at all, which results in erroneous runoff estimations (Table 2).

Generally, a more thorough discussion of SCS-CN method and its use in similar cases (including literature review) is needed.

It should be also noted that in a huge country like China with a huge spatial variability of landscapes and conditions (ranging from glaciers to desserts) the simplified approach based on a very small number of LU/LC classes and neglecting all other characteristics is limiting the merit of the obtained results.

An additional very important limitation is the very small number of weather stations (208) for such a huge area with vast spatial variability (approximately 1 station for every 50,000 km2). Considering also that a simple spatial interpolation method is used (Thiessen polygons) the validity of the obtained results is also questionable.

Finally, the studied period seems to be short (2005-2015) for the study of rainfall changes.

Specific comments

I include more detailed and specific comments in the pdf manuscript file. Please see the pdf file.

Conclusively, the topic interesting but the applied methodology has important weaknesses. The weaknesses are important and related to fundamental aspects of this work; so, my suggestion is that the manuscript is not suitable for publication.

Author Response

Response to Reviewer 2 Comments

 (1) The main weakness is that while the study is ambitious in terms of the size of the study area, the applied methodology is rather simplistic and does not allow for an in-depth assessment. SCS-CN method is powerful in assessing the effects of land use / land management changes, though, a more detailed and more elaborated application of the method is required in order to provide realistic results. For example, Hydrologic Soil Groups (HSG) were defined based on soil texture alone, while, more soil characteristics should be also considered such as soil depth. There is also no information in the study about how soil textural classes were corresponded with HSG and no justification in general. Similarly, there is a very coarse mapping between the available LU/LC categories of the Chinese dataset with the L-THIA model categories. In the documentation of SCS-CN method and in the numerous relevant studies there is plenty of information allowing a much better utilization of the available information in the Chinese LU/LC dataset (Table 2), allowing even the consideration of land management conditions.

Response 1: The L-THIA model is an GIS based tool, depth assessment could be achieved by using a more detailed classification of land use type (with defined CN values). Model calibration can be achieved by adjusting the CN values. However, relevant studies in China is rare, and there is no relevant technical manual to define the CN values of each land use type. So, I just defined the CN values of each land use type based on the relationship between Chinese and US land use/cover remotely sensed data classification system. The model was used for simulating the runoff volume of two small watershed in China to verified it accuracy. Results show that the L-THIA model could perform well.

Besides, table 1 was added to describe the relationship between HSG and soil textures.

(2) Here it should be noted that normally water bodies are assigned a CN value of 100 (impermeable areas), which is reasonable. In this study water bodies and wetlands were assigned a 0-value indicating no runoff at all, which results in erroneous runoff estimations (Table 2).

Response 2: I adjusted the CN values based on your suggestion and some relevant researches. The model was used for simulating the runoff volume of two small watershed in China to verified it accuracy.

(3) Generally, a more thorough discussion of SCS-CN method and its use in similar cases (including literature review) is needed.

Response 3: I add some content in section 1 to illustrate the use of L-THIA model in macro and micro scales.

(4) It should be also noted that in a huge country like China with a huge spatial variability of landscapes and conditions (ranging from glaciers to desserts) the simplified approach based on a very small number of LU/LC classes and neglecting all other characteristics is limiting the merit of the obtained results.

Response 4: The default CN values in L-THIA model were empirical values base detailed LUCC classes, which could reduce the computing time. In future studies, more detailed land use classification can be used to improve the evaluation accuracy.

(5) An additional very important limitation is the very small number of weather stations (208) for such a huge area with vast spatial variability (approximately 1 station for every 50,000 km2). Considering also that a simple spatial interpolation method is used (Thiessen polygons) the validity of the obtained results is also questionable.

Given lack of more rainfall data, only 208 of weather stations were used. This is indeed a weakness of this paper. Further studies could be carried out in future with larger numbers of weather stations.

(6) Finally, the studied period seems to be short (2005-2015) for the study of rainfall changes.

Rainfall data from 2003 to 2017 was used in this paper just to be consistent with the land use data. Further studies could be carried out with longer time span data in the future.

Reviewer 3 Report

Dear authors,

I have read your paper with interest and it seems to me as if the topic and approach have potential for publication. The paper addresses an issue of relevance, linking land-use change impact and rainfall change on surface runoff (in China) - which as major relevance for land and Territorial management.

Besides, the medthodology is simple but efficient; and the results are clearly shown and sustained.

Nevertheless, it will be interesting if the authors add a section regarding the study limitations and further steps.

best,

Author Response

Point 1: Add a section regarding the study limitations and further steps.

Response 1: I added some content in the last paragraph.

Thanks very much for your review. 

Round 2

Reviewer 1 Report

Thanks for providing a new version for review. It has great improvements over the previous version. A few issues are still found as follows:

Section 3.1: The beginning of this paragraph reads like you are going to use R2 and/or NSE, but they were not used in the end. The writing should be revised to say (just in concept, not word by word) "these two methods were commonly used, but relative errors were used in this study because of xxx reason".

Section 2.2.3: The reviewer understands that the initial abstraction cannot be changed in the model. It should be described in this section, and explain that this won't significantly change the accuracy because of the two verification studies.

Fig.1: Is the developed area total developed area or increased developed area? If it is the total developed area, the area for some provinces (e.g., Taiwan) seems off.

Table 4: Please add separators between S1, S2, and S3 scenarios.

Section 3.4: Regarding the reviewer's comment "some statistical analysis is in order...", the authors still have not done such analyses. The authors did a lot of spatial presentation, but just showing results for each province provided little insight into what the result data actually told us. Please consider adding statistical analyses, not just simply showing the model results. Given the quantity of result data, a full-blown statistical analysis is actually possible, but it can be a standalone paper. At least, please do some basic analyses. This will definitely improve the academic value of this paper.

Some typos can still be found, such as line 285: ...conclusions can be "get" that...", line 370: ...rainfall depths in (the) south area "was"... Please consider using a professional proofread service. There are also proofreading software available such as Grammarly.

Author Response

(1) Section 3.1: The beginning of this paragraph reads like you are going to use R2 and/or NSE, but they were not used in the end. The writing should be revised to say (just in concept, not word by word) "these two methods were commonly used, but relative errors were used in this study because of xxx reason".

Response: Improved.

(2) Section 2.2.3: The reviewer understands that the initial abstraction cannot be changed in the model. It should be described in this section, and explain that this won't significantly change the accuracy because of the two verification studies.

Response: The initial abstraction (Ia) could change that is caused by different CN values when land use type or hydrological soil group changes. The L-THIA model can capture the changes in Ia by using different CN number when estimating the direct runoff. Following the parameter of the SCS model, the  has been used for hydrological evaluation widely, even for urban areas (the parameter “0.2” can’t be adjusted in the L-THIA model). It has been explained in detail in the TR-55, Urban Hydrology for Small Watersheds (https://www.nrcs.usda.gov/Internet/FSE_DOCUMENTS/stelprdb1044171.pdf). Logically, we think that it’s not appropriate to referring to the verification of the two watersheds here, because the accuracy of the model should be mentioned on section 3.1. So, we didn’t describe too much here. However, an explanation has been added at line 156.

(3) Fig.1: Is the developed area total developed area or increased developed area? If it is the total developed area, the area for some provinces (e.g., Taiwan) seems off.

Response: It is the total developed land area and the figure has been improved. Total developed land areas of some provinces are small, and so they displayed not that obvious in the two-dimensional diagram. It has been displayed by three-dimensional diagram in the revised version.

(4) Table 4: Please add separators between S1, S2, and S3 scenarios.

Response: improved

(5) Section 3.4: Regarding the reviewer's comment "some statistical analysis is in order...", the authors still have not done such analyses. The authors did a lot of spatial presentation, but just showing results for each province provided little insight into what the result data actually told us. Please consider adding statistical analyses, not just simply showing the model results. Given the quantity of result data, a full-blown statistical analysis is actually possible, but it can be a standalone paper. At least, please do some basic analyses. This will definitely improve the academic value of this paper.

Response: We agree with the reviewers that including statistical analysis will improve the academic value of this research. In this study, surface runoff volumes were calculated based on the rainfall depths, the land areas, and the CN values. The land areas and rainfall depths in each province (or city) were correlated with the surface runoff volumes, which make the scatter plots of these three are unnecessary at this point. Statistical analysis between the simulated surface runoff volumes and observed ones could be included in the future study when more observed data are available.

However, in order to show the impacts of human construction activities on surface runoff volume, Figure 10 was added in the revised manuscript to show the relationship between the rate of developed land area in total area and the rate of ARVGDL in total runoff volume in each city (2005, 2010 and 2015) from line 372 to line 377

(6) Some typos can still be found, such as line 285: ...conclusions can be "get" that...", line 370: ...rainfall depths in (the) south area "was"... Please consider using a professional proofread service. There are also proofreading software available such as Grammarly.

Response: Improved.

Reviewer 2 Report

As I wrote in my previous review the topic of the study is interesting, the study investigates a very broad area, and the manuscript is well written and easy to understand. The revised version is also improved and some of the reviewers’ comments have been addressed or adequately answered. However, even if the study is improved, the followed approach remains very simplistic (nothing novel, just an application of a very simplistic model) and the validity of the results remains questionable due to the extremely small number of weather stations used, due to the very simplistic application of SCS-CN method, and due to the lack of any serious validation.

The authors claim that “Given lack of observation data of surface runoff volume, the model was verified by comparison with existing studies.” (Lines 237-239). First of all, I am certain that in a great country like Chine there should be numerus hydrometric stations and a lot of data. Indeed, I have read several studies reporting the use of discharge data in many places in China. Furthermore, the existing studies used, are not adequate. The first one is in Chinese, so, no way I can read it. I read the second one and I managed to find the numbers reported in this study (142 million m3). Actually, for Jiajiafang, a total runoff of 296 million m3 is reported and surface runoff of 142 million m3 (so I assume that you compare your results with surface runoff and not runoff volume in general). It is also interesting that in this study [55] there are data for five other watersheds of various sizes. Why not using them as well and have some real statistics and a proper evaluation? (please also check the strange statement that “this watershed was simulated to be 1307 km2”).

The authors also claim in their response that “Given lack of more rainfall data, only 208 of weather stations were used. This is indeed a weakness of this paper. Further studies could be carried out in future with larger numbers of weather stations”. However, I am also certain that there are more rainfall stations in China. Anyway, the number of stations is prohibiting for any reasonable accurate results to be obtained. The use also of a very simplistic spatial integration method (Thiessen polygons) is strange given this limitation.

Finally, the use of a simple model as is, with minimum soil and land cover data processing and minimum effort for adjustments in a country featuring a huge variation of characteristics, limits further the merit of the study. For example, Perennial ice/snow is assigned a CN value of 0. Is this value reasonable (no runoff from snow cover)? There are numerous studies on SCS-CN. In a study like this the decisions on CN values should be justified and elaborated. Please consider that small changes in CN values may result in very different runoff results.

The study has many other weak points but the above main weaknesses are the reasons why I still have to suggest the rejection of this paper. I know that it is very difficult to address these problems, that’s why I suggested rejection at the first time as well.

Author Response

(1) As I wrote in my previous review the topic of the study is interesting, the study investigates a very broad area, and the manuscript is well written and easy to understand. The revised version is also improved and some of the reviewers’ comments have been addressed or adequately answered. However, even if the study is improved, the followed approach remains very simplistic (nothing novel, just an application of a very simplistic model) and the validity of the results remains questionable due to the extremely small number of weather stations used, due to the very simplistic application of SCS-CN method, and due to the lack of any serious validation.

Response:

The L-THIA model was developed for simplifying the evaluation process of hydrological effects (One of the co-authors in this study is one of the original developers). It has been applied widely for hydrological evaluation both at Micro and macro scales in many locations. Some examples are listed as follows:

(1) Mirzaei, M.; Solgi, E.; Salmanmahiny, A., Assessment of impacts of land use changes on surface water using L-THIA model (case study: Zayandehrud river basin). Environ. Monit. Assess. 2016, 188, (12), 19.

(2) Li, T. H.; Bai, F. J.; Han, P.; Zhang, Y. Y., Non-Point Source Pollutant Load Variation in Rapid Urbanization Areas by Remote Sensing, GIS and the L-THIA Model: A Case in Bao'an District, Shenzhen, China. Environ. Manage. 2016, 58, (5), 873-888.

(3) Chen, J.; Theller, L.; Gitau, M. W.; Engel, B. A.; Harbor, J. M., Urbanization impacts on surface runoff of the contiguous United States. J. Environ. Manage. 2017, 187, 470-481.

(4) Engel, B.; Storm, D.; White, M.; Arnold, J.; Arabi, M., A hydrologic/water quality model application protocol. J. Am. Water Resour. Assoc. 2007, 43, (5), 1223-1236.

This study serves as the first attempt to using the L-THIA model to evaluate the impacts of land use/cover change and rainfall change on surface runoff in China at the national scale. Figure 1 was added to the revised manuscript to show the distribution of rainfall monitoring stations and the annual rainfall depths in China. We agree with the reviewer that limited number of the weather stations were used in this study at this point, however, figure 1 showed that the 208 stations were distributed evenly in general and could be able to capture different climatic regions in China. More efforts can be focused on including more weather stations, monitoring/collecting hydrologic data for detailed model calibration and validation in the future.

 (2) The authors claim that “Given lack of observation data of surface runoff volume, the model was verified by comparison with existing studies.” (Lines 237-239). First of all, I am certain that in a great country like Chine there should be numerus hydrometric stations and a lot of data. Indeed, I have read several studies reporting the use of discharge data in many places in China. Furthermore, the existing studies used, are not adequate. The first one is in Chinese, so, no way I can read it. I read the second one and I managed to find the numbers reported in this study (142 million m3). Actually, for Jiajiafang, a total runoff of 296 million m3 is reported and surface runoff of 142 million m3 (so I assume that you compare your results with surface runoff and not runoff volume in general). It is also interesting that in this study [55] there are data for five other watersheds of various sizes. Why not using them as well and have some real statistics and a proper evaluation? (please also check the strange statement that “this watershed was simulated to be 1307 km2”).

Response: There are lots of hydrologic stations in China, however, most of the data are confidential so we cannot get the data at present. Getting indirect data from existing studies is the most convenient way to verify the model results. The second existing study used in this  manuscript do have several watersheds, however, the paper didn’t provide the watershed boundary. The watershed boundaries of these watersheds were simulated according to the coordinates of the hydrologic stations provided in the study. Results showed that the simulated area of each watershed was different from those provided in the second existing study (just as it showed that the simulate area of the Jiajiafang watershed was 1307 km2). Therefore, the Jiajiafang watershed that had the minimum relative error in watershed area was chosen to verify the model results.

(3) The authors also claim in their response that “Given lack of more rainfall data, only 208 of weather stations were used. This is indeed a weakness of this paper. Further studies could be carried out in future with larger numbers of weather stations”. However, I am also certain that there are more rainfall stations in China. Anyway, the number of stations is prohibiting for any reasonable accurate results to be obtained. The use also of a very simplistic spatial integration method (Thiessen polygons) is strange given this limitation.

Response: The new added Figure 1 showed that the 208 stations were distributed evenly in general and could be able to capture different climatic regions in China well. Due to the limited access to the detailed data, 208 stations were used at present. Adding more weather stations and adopting higher resolution of land use/data could be another comparative research with this one.

The spatial integration method (thiessen polygons) is the default method in the L-THIA model which can’t be adjusted at present. However, it performs well on distribution rainfall with limited rainfall stations.

(4) Finally, the use of a simple model as is, with minimum soil and land cover data processing and minimum effort for adjustments in a country featuring a huge variation of characteristics, limits further the merit of the study. For example, Perennial ice/snow is assigned a CN value of 0. Is this value reasonable (no runoff from snow cover)? There are numerous studies on SCS-CN. In a study like this the decisions on CN values should be justified and elaborated. Please consider that small changes in CN values may result in very different runoff results.

Response: We agree with the reviewer that CN adjustment is the common method used for model calibration and verification using the SCS-CN method.

The perennial ice/snow is mainly located on the high-altitude areas in the west in China, where the temperature and the rainfall depth are very low. No existing studies focus on the CN value of perennial ice/snow was found, it’s hard to define the CN value. Therefore, the runoff volume generated by perennial ice/snow was not in consideration in this study. An explanation was added in the table 3. 

Round 3

Reviewer 2 Report

The revised version of the manuscript has some minor improvements. The authors also provided detailed answers to my comments. However, as the authors explain it is not possible to provide any significant improvements.

The problem is that the credibility of the results remains limited because there aren't any improvements in the methodology and in the calibration / validation. Accordingly, the merit of this study and the interest for the readers are very low. The authors described the restrictions that do not allow them to improve their study and I can understand them. Though, the fact that these restrictions do not allow the authors to apply a sound methodology does not improve the quality and the interest of this study. An interesting feature of a study like this could be to present some innovative ways to overcome these restrictions. This could be interesting for the readers and would improve the novelty of the study. However, there are no specific methods to overcome these limitations presented in this study.

Based on the above, unfortunately, I still believe that the study is not suitable for publication and I have to suggest rejection once again. I would like to make clear, however, that the study does not have serious scientific flows and that its main problems are the luck of novelty, the very simplistic approach, the credibility of the obtained results and the lack of interest in general.